# Morphological disparity across the K-Pg boundary in mollusk shells: A theoretical morphology approach

Gabriela Contreras-Figueroa[1]☯, Austin J. W. Hendy[2]☯, José L. Aragón🄳[1]☯*

**1** Universidad Nacional Autónoma de México, Centro de Física Aplicada y Tecnología Avanzada, Querétaro, Mexico, **2** Natural History Museum of Los Angeles County, Los Angeles, California, United States of America

☯ These authors contributed equally to this work.
* jlaragon@unam.mx

## Abstract

Theoretical morphospace as a tool for understanding evolutionary changes in molluscan shell form is applied to analysis of biodiversity change across the Cretaceous-Paleogene extinction event. Although empirical morphospaces based on geometric morphometric descriptors are widely used, theoretical morphospaces offer better insights into the macroevolutionary constraints on morphospace occupation. The purpose of this work is to characterize the morphological disparity in mollusk shells during the Cretaceous and Paleogene, and specifically across the Cretaceous-Paleogene (K-Pg) boundary. Disparity measurements are derived from a recently proposed theoretical morphospace that is based on shell geometric parameters closely related to molluscan ecology and functional morphology. The case study focuses on representative families of the classes Bivalvia and Gastropoda, during the Late Cretaceous and early Paleogene of the Northeast Pacific coast, spanning from Washington to the Baja California Peninsula. The relationship between shell morphology and mode of life in mollusks and how geometric models can be used to understand shell morphology and disparity are discussed. Mapping bivalves and gastropods in morphospace across the K-Pg boundary reveals a marked loss of morphological disparity immediately after the extinction event. This was followed by a recovery during the Paleocene, particularly among infaunal and mobile taxa, where increased mobility and deposit-feeding strategies proved advantageous in low-nutrient environments. Applying the model to future research could contribute to the understanding of the paleoecological and ecomorphological factors that have influenced the diversity of these groups. We also note that the scarcity of disparity studies related to mollusks across the K-Pg boundary is striking and calls for more research in this area.

**Data availability statement:** All relevant data are within the manuscript and its Supporting Information files.

**Funding:** Student Collections Study Award and Paleontological Research Institution for PRI John W. Wells Grants-in-Aid of Research Program. The funders had no role in study design, data collection and analysis, decision to publish, or preparation of the manuscript.

**Competing interests:** The authors have declared that no competing interests exist.

## Introduction

The phylum Mollusca is one of the most diverse animal groups, exhibiting remarkable morphological disparity and a variety of modes of life that have led them to inhabit different ecological niches. Mollusk shells with a wide diversity in shape, and an abundant and complete fossil record [1,2] offer an exceptional model for studying evolutionary patterns such as extinctions and community recovery. Therefore, morphological studies focused on quantifying the diversity of mollusk shell forms are important for understanding these processes [3–5].

In this context, morphological disparity has emerged as a paleobiological concept for characterizing evolutionary changes in body plans [6]. In practice, disparity is commonly visualized as the position of taxa within a given morphospace, defined by a set of morphological descriptors that vary depending on the approach and method of acquisition [7,8]. Using this approach, disparity is quantified based on the amount of space occupied and the position of a taxon relative to others [9]. Various aspects of morphological variability across geological time can be captured by employing diverse disparity measures. For example, these analyses can detect morphological trends such as increased clustering, replacement, and successive diversification of taxa occupying vacated morphological space, or the rapid filling of morphospace during evolutionary radiations [10–13]. This type of study requires a series of morphospaces corresponding to specific temporal intervals and limited taxonomic breadth.

The majority of disparity analyses use empirical morphospaces based on geometric morphometric descriptors [14], and are generated by statistical methods for reducing variables, such as principal component analysis (PCA). However, PCA-based morphospaces can be difficult to interpret, as they are influenced by sample size and abstract axes [9,14]. Alternatively, theoretical morphospaces define shape variation through mathematical models independent of sampling, allowing the exploration of both existing and non-existent forms [15]. Despite the numerous geometric models proposed to characterize the morphology of mollusk shells, their application in disparity analyses has been quite limited in this group. However, some studies have successfully applied empirical and/or theoretical morphospace approaches to mollusks [16–18]. For instance, analyses of veneroid bivalves across the K-Pg boundary revealed unoccupied morphospace regions [19] and a post-extinction shift toward deeper-burrowing shell forms [3].

Theoretical models are particularly valuable in mollusk shell studies, as many shell forms can be described by relatively simple geometric parameters, such as those based on logarithmic spirals [20–22]. Since Raup's pioneering work [22–26], numerous models have been proposed, though practical challenges remain—especially in quantifying theoretical parameters in real specimens. Recent methodological advances have begun to address these issues, offering standardized ways to quantify a wide range of shell morphologies [27–29], especially standardized methodology for quantifying the same geometric parameters across various shell morphologies including planispiral, helicoidal, valve, and conic shapes [30].

Establishing a direct relationship between an organism's form and function does not always yield a clear adaptive significance. Despite this, several studies have

contributed to the understanding of shell morphology and mode of life in mollusks from a broad perspective. These investigations reveal how morphological variations are linked to functional characteristics, allowing mollusks to adapt to specific movements and diverse feeding strategies [31–39]. These links are particularly important when assessing response to major evolutionary events such as mass extinctions.

Extinctions stand out among macroevolutionary processes because of their crucial role in modifying and abruptly interrupting evolutionary trajectories [40], and they are often accompanied by geological or environmental changes that reflect the severity of the triggering events [41]. From a biological perspective, mass extinctions have primarily been documented through changes in taxonomic diversity, but they can also be inferred from shifts in ecological structure [42] or biogeographic patterns [43]. A less common approach involves characterizing changes in morphological disparity, where extinctions are observed as empty areas or changes in taxa distribution in the morphospace [44,45]. As the distribution of species within a morphospace reflects their functional traits and ecological roles, changes in this distribution can be used to reconstruct ecological release or niches. Evaluating morphological and ecological disparity is an important tool for understanding biodiversity dynamics, particularly during mass extinction events [46].

The Cretaceous–Paleogene (K-Pg) boundary represents one of the most dramatic examples of such evolutionary disruption, caused by the Chicxulub impact approximately 66 million years ago [47–50]. This event triggered global environmental consequences, including the collapse of primary productivity due to global darkness and ocean acidification [51–53], which occurred heterogeneously across the planet, suggesting spatially and temporally variable effects of the K-Pg boundary event [54]. Mollusks were profoundly affected, a clear decline in the number of mollusk taxa—at the family, genus, and species levels—is observed immediately following the extinction event. The diversity of mollusks in the Eastern Pacific is summarized in Fig 1 and clearly illustrates this taxonomic decline during the Danian, a period considered to represent an interval of ecological recovery (data from [55]). After this phase, a significant recovery is recorded during the Selandian, when species richness even surpasses Maastrichtian. However, beyond taxonomic diversity, it is also crucial to assess how extinction events influence morphological disparity and ecological strategies.

Although extinction at the end of the Cretaceous is a well-studied macroevolutionary event, there is a knowledge gap regarding the morphological disparity of molluscan shells compared to other taxonomic groups, for instance the vertebrates [56–58]. The study of mollusks in an ecomorphological context is still limited, despite the potential correlations in their ecomorphology and the large number of specimens available in paleontological collections. The purpose of this work is to apply

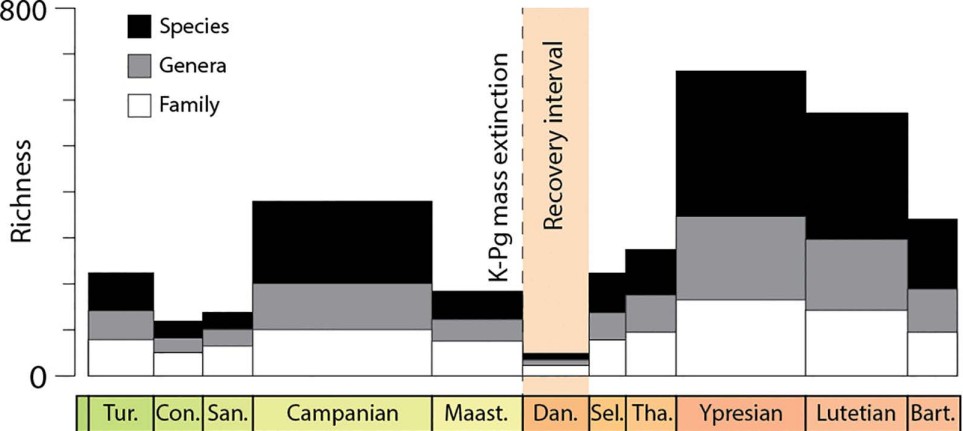

**Fig 1. Late Cretaceous–Paleocene changes in molluscan faunas of the Eastern Pacific.** Diversity at the family, genus, and species levels (measured using the range-through method), based on data from [55].

a recently proposed theoretical morphospace analysis to characterize the morphological disparity in mollusk shells across the K-Pg boundary based on shell geometric parameters that can be related to their ecomorphology. The case study focuses on representative families of the classes Bivalvia and Gastropoda, before and after the Cretaceous-Paleogene (K-Pg) boundary on the Northeast Pacific coast, spanning from British Columbia to the Baja California Peninsula. Deposits in this area offer a rich diversity of mollusk fauna associated with warm and shallow-water environments [59].

This study applies disparity measures across theoretical morphospaces from the Cretaceous to the Eocene to examine unoccupied areas and shifts in morphological distribution across the K-Pg boundary. These changes, linked to variations in locomotion, life position, and morphology, suggest an ecological response to nutrient-poor environments following the extinction event, favoring infaunal and actively mobile taxa. These analyses demonstrate the utility of this morphospace framework and its geometric parameters for exploring mollusk shell ecomorphology during extinction events.

## Methods

### Fossil material

The fossil material used in this study comprised 547 specimens from the classes Bivalvia (270 specimens from 22 families) and Gastropoda (277 specimens from 19 families). Specimens were selected from the following locations along the North American Pacific coast: Baja California, California, Oregon, Washington, and British Columbia. This selection covers a geological range extending from the Late Cretaceous (Campanian) to the Eocene (Priabonian). The families and species were selected using the taxonomic resource developed for the National Science Foundation-funded Cretaceous Seas of California, Connecting Cretaceous Seas, and Eastern Pacific Invertebrate Communities of the Cenozoic (EPICC) digitization projects [60]. This resource provides accessible digital data on species taxonomy, stratigraphic range, geographic distribution, body size, and ecological traits (S1 Table). In the present study, only one exemplar specimen per species from the selected families was used; therefore, the general morphology of the species was studied without considering intraspecific differences.

Intraspecific variation is anticipated to be minimal in the context of shape differences across families and classes. Although intraspecific variation is relevant for understanding morphological diversity, it is generally lower than the variation observed among species, genera and families. The model used in this study was designed to capture macroevolutionary patterns in the overall shell morphology of mollusks; therefore, small intraspecific differences, particularly those related to ornamentation, may not be reliably detected. Furthermore, given the geological timescales considered, the analysis focused on morphological divergence among higher-level lineages rather than intraspecific variation. Whenever possible, specimens from type collections (holotypes, paratypes, or hypotypes) were selected to ensure an accurate representation of each species. Because type collections are typically based on adult specimens, they provide the most reliable material for taxonomic and morphological comparisons. From these, we selected the best-preserved and most complete fossils available.

Digital images of fossil specimens were required for morphological analysis. The guidelines of Callomon [61] were followed to ensure proper alignment of the shells from the apertural, lateral, and apical views. Photographs were obtained from various sources: Invertebrate Paleontology Department of the Natural History Museum of Los Angeles (LACMIP), Paleontological Research Institution (PRI), University of California Museum of Paleontology (UCMP), and iDigBio (Integrated Digitized Biocollections), as well as images from bibliographic reviews [62–65]. The list of reviewed specimens is presented in the supplementary information (S1 Table), indicating the sources of the photographs. No permits were required for the described study, which complied with all relevant regulations.

### Measures and theoretical morphospace

The geometric parameters of the shell were measured based on its spiral geometry, following the mathematical approach and methodology previously proposed by [30]. The morphological characteristics are associated with parameters that describe both the structure of the logarithmic spiral within a fixed coordinate system *x-y-z*, as well as the generating curve (shell aperture) that moves along this spiral following the Frenet reference frame (tangent-normal-binormal).

This model proposed six parameters for the three-dimensional modeling of shells: whorl expansion rate (*b*), horizontal distance of the spiral to the coiling axis (*d*), vertical displacement of the spiral along the coiling axis (*z*), elliptical shape of the aperture (*a*), displacement of the aperture in relation to the coiling axis (*k*), and rotation angles of the aperture with respect to the coiling axis (*ϕ*). These parameters were obtained for all specimens to generate a three-dimensional model of their morphology; however, only the two parameters that best captured the relationship between shell shape and ecological type were used for morphospace analysis: $k − ϕ$ for bivalves and $z − a$ for gastropods (see next subsection about ecomorphology). Fig 2 shows the measurements required to obtain these parameters in the specimen photographs, as well as the variation of the shell shape when the corresponding parameter value is changed.

Two-dimensional theoretical morphospaces were constructed for each class by systematically varying selected parameters. Specimens were mapped within these morphospaces based on taxonomic and ecological context (S1 Table). The stratigraphy of the sampled groups covers from 100.5 million years ago (Ma) to 33.9 Ma. These periods included the Late Cretaceous (Cenomanian, Turonian, Coniacian, Santonian, Campanian, Maastrichtian), Paleogene (Danian, Selandian, Thanetian), and Eocene (Ypresian, Lutetian, Bartonian, and Priabonian). For the morphospace analysis these ages were grouped into six time periods: Time 1 (100.5–83.6 Ma; Cenomanian-Santonian), Time 2 (83.6−66 Ma; Campanian-Maastrichtian), Time 3 (66–61.6 Ma; Danian), Time 4 (61.6−56 Ma; Selandian-Thanetian), Time 5 (56–47.8 Ma; Ypresian), and Time 6 (47.8–33.9 Ma; Lutetian, Bartonian, Priabonian). Models and theoretical morphospaces were constructed using *Wolfram Mathematica* [66].

## Ecomorphology

Ecological information on bivalves and gastropods focused on the lifestyle or mode of life. Ecological categories were divided based on their tiering (epifaunal, semi-infaunal, and infaunal) and mobility (stationary, facultatively mobile, or

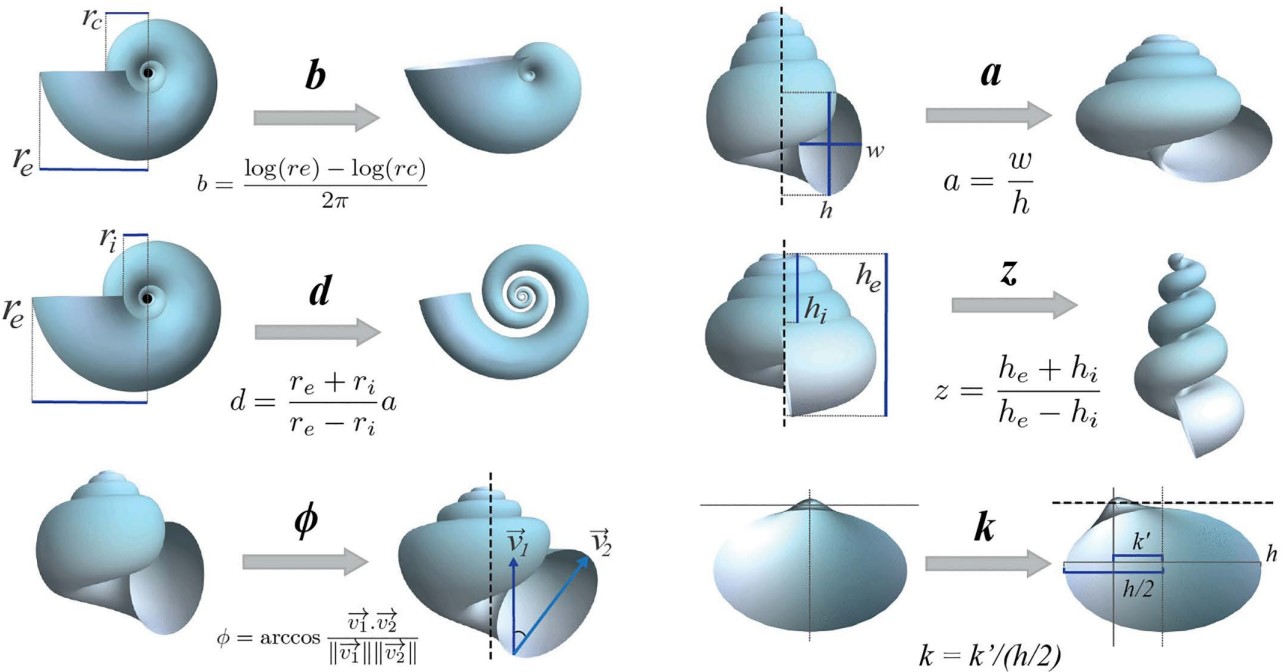

**Fig 2. Measurements and equations required to calculate parameters in specimen photographs.** The theoretical modification of the shell is shown when the value of the parameter is increased. Dashed lines represent the coiling axes. For further details on the measurements, see Ref. [30].

creeping). Bivalves were grouped into four ecological categories: stationary epifauna, facultatively mobile semi-infauna, facultatively mobile infauna, and deep infauna. Similarly, gastropods were categorized as facultatively mobile epifauna, creeping epifauna, facultatively mobile infauna, and creeping infauna [60].

All the parameters together reconstruct the three-dimensional morphology of the shell, allowing the reconstruction of the structures that shape its ecological strategies. However, this study focuses on the use of two parameters for each mollusk group, following the ecomorphological references that indicate the most relevant relationships. In addition, selecting these specific parameters provides better visualization and understanding when generating the morphospace.

The relationship between morphological parameters and ecological strategies (tiering and mode of life) is summarized in Table 1. The bivalves were characterized according to parameters that defined the rotation ($\phi$) and displacement ($k$) of the aperture in relation to the coiling axis located in the umbo region. These morphological characteristics define the reduction or expansion of the anterior region of the shell, which is associated with the type of locomotion exhibited by organisms adhering to the substrate or excavating, respectively [31,34,67].

Parameters defining the gastropod shells in this study were related to the vertical displacement of the spiral along the coiling axis ($z$) and the shape of the aperture associated with an ellipse ($a$). These parameters correlate with variations in spire height, folding in the columella, and laterally flattened apertures in burrowing organisms, but for sedentary organisms, folds may be absent, providing stability when attached to substrates [34,38,39].

## Disparity measurements

Morphological disparity across the K-Pg boundary was analyzed through examining the way specimens of bivalves and gastropods were distributed in theoretical morphospaces over the defined geological periods. Disparity was quantified using two complementary measures: the area delimited by the convex hull and the Mean Pairwise Distance (MPD). The convex hull of a set of points $X$, formally defined as the smallest convex set that contains $X$, represents the total extension

**Table 1. Morphological parameters associated with mode of life in bivalves and gastropods. Information based on references [31,34,38,39,67].**

|  | Mode of life | Parameters | Anatomical implications | Function |
|---|---|---|---|---|
| **BIVALVIA** | Epifauna | High values of $k$ (aperture displacement in relation to coiling axis). | These parameters facilitate development of the posterior region and the formation of the byssal retractor muscles. | This morphology plays an important role in attachment to stationary epifauna. |
|  |  | High values of $\phi$ (aperture rotation with respect to the coiling axis). |  |  |
|  | Infauna, semi-infauna or deep infauna | Low to moderate value of $k$ (aperture displacement in relation to coiling axis). | Both parameters, especially $\phi$ rotation, allow for an anterior area closer to the umbo and can be anatomically associated with an elongated anterior region, resulting in foot enlargement and a triangular anterior edge. | These characteristics facilitate easy penetration into the sediment and subsequent burial. |
|  |  | Moderate values of $\phi$ (aperture rotation respect to the coiling axis). |  |  |
| **GASTROPODA** | Epifauna | Moderate to high values of $a$ (aperture shape circular to width/opened). | These forms exhibit tight coiling and a low spire, with columellar folds often absent. The shell aperture, nearly circular in outline, reflects the shape of the foot— a feature also characteristic of sedentary organisms and limpet-like shells. | This morphology provides stability when attached to substrates through strong pedal adhesion. It has also been proposed as resistant to shell crushing. |
|  |  | Low values of $z$ (low vertical displacement of spire). |  |  |
|  | Semi-infaunal | Low to moderate values of $a$ (elongated aperture). | These parameters correlate with variations in spire height, folding in the columella, and laterally flattened flanks due to an elongated aperture. | These features have been correlated with the burial process. They improve the attachment of the columellar muscles. Also, these gastropods have smooth flanks, which help reduce drag. |
|  |  | Moderate to high values of $z$ (high vertical displacement of spire). |  |  |

of occupied morphospace for each ecological category. Its area was calculated using the *ConvexHullMesh* function in *Wolfram Mathematica* [66]. In contrast, the MPD characterizes the distribution of points within this region, and was estimated using the *DistanceMatrix* function, which computes all pairwise distances between elements mapped in the morphospace. These disparity metrics were applied to each ecological category of Bivalvia and Gastropoda, and their values were graphically represented across the established geological timeframes.

## Results

Some examples of three-dimensional models obtained by measuring all geometric parameters are illustrated in Fig 3. Notably, the similarity between the fossil specimens and the theoretical model is evident, implying that the parameters have the potential to describe the morphology of various shell types. However, certain morphological features, such as the siphonal canal or non-elliptical aperture, cannot be fully replicated due to the limitations of the geometric model.

The results of the disparity analysis were conducted independently for Bivalvia and Gastropoda, and they are presented in the following order: i) theoretical morphospace by systematic variation in the geometric parameters; ii) mapping morphospace with the specimen distribution (convex hulls) by ecological category across the established geological time; iii) disparity measure graphs including convex hull area and MPD by time; iv) morphospace with taxonomic influence of families on the distribution of ecological categories during the Campanian-Maastrichtian (Time 2) and Paleocene (Time 3–4).

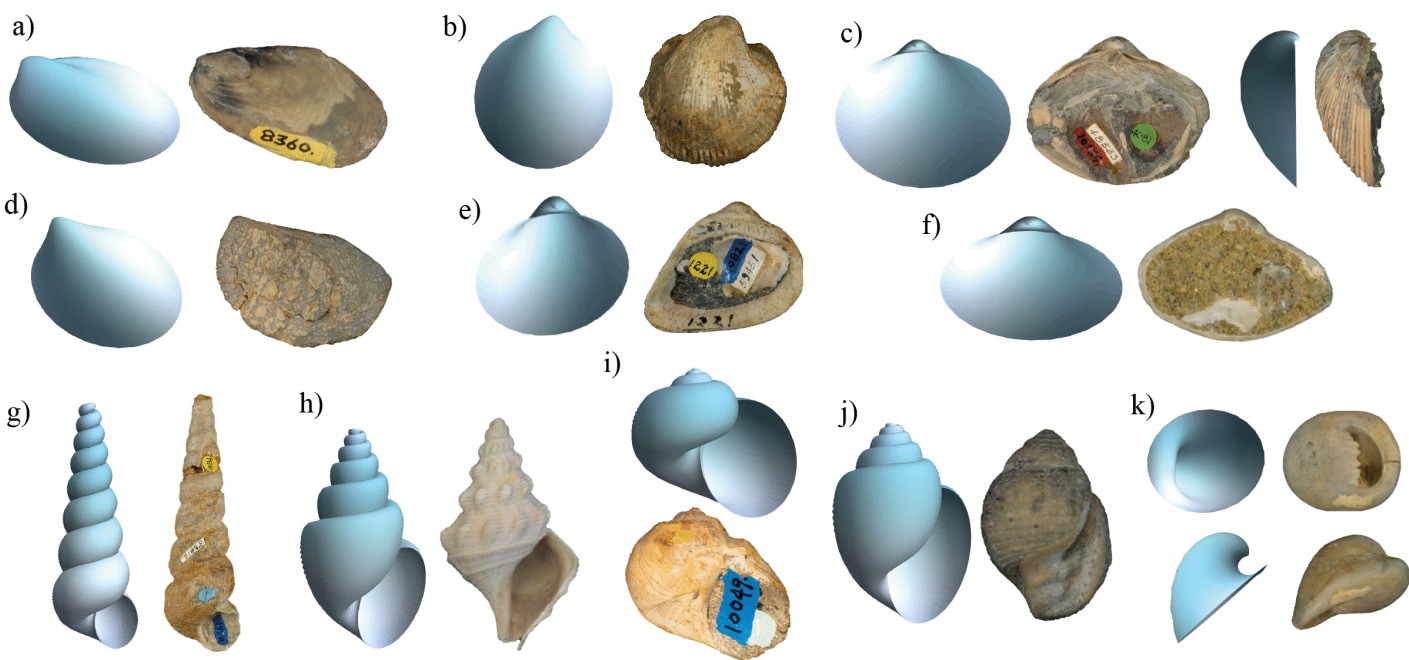

**Fig 3. Examples of digitized bivalve and gastropod specimens and their respective models obtained using geometric parameters: a)** *Bathymodiolus willapaensis* (LACMIP 5802 Type 8360; Mytilidae); **b)** *Agnocardia sorrentoensis* (LACMIP 22312 Type 12752; Cardiidae); **c)** *Cymbophora bella* (LACMIP 24081 Type 10154; Mactridae); **d)** *Yaadia robusta* (LACMIP 26298 Type 10193; Steinmanellidae); **e)** *Cucullaea (Idonearca) melhaseana* (LACMIP 10738 Type 10821; Cucullaeidae); **f)** *Caryocorbula dickersoni* (LACMIP 16115 Type 6592; Corbulidae); **g)** T*urritella chaneyi* (LACMIP 10659−1; Turritellidae); **h)** *Goniocheila wilsoni* (LACMIP 17535−6; Aporrhaidae); **i)** *Gyrodes quercus* (LACMIP 24106−19; Naticidae); **j)** *Ravinella lipmanorum* (LACMIP 41691−16 Type 14724; Acteonidae); **k)** *Velates vizcainoensis* (LACMIP 27083−11 Type 8875; Neritidae).

## Bivalves

Most bivalve specimens clustered in a region of the $k-\phi$ morphospace (Fig 4) with a relatively uniform morphology. However, when the distribution of bivalves within the morphospace is separated according to ecological category and time interval, notable differences are observed. During the Danian (Time 3), there was a major decline in the disparity of all ecomorphologies (Fig 5). However, as shown in Fig 5c, organisms with the capacity for burrowing and active locomotion (facultatively mobile infauna) were the only ones that did not experience a decline according to the MPD measure, or exhibited only a slight decrease following the convex hull measure. Over the following 10 million years (Time 4), the paleoecological recovery period was characterized by faunal turnover. During this interval, all categories regained disparity levels comparable to those prior to the extinction event, except for the facultatively mobile semi-infaunal group, in which only Glycymerididae persisted across the K-Pg boundary (Fig 6).

**Stationary epifauna.** This category has a lower species richness (Time 1 = 13 spp.; Time 2 = 19 spp.; Time 3 = 1 sp.; Time 4 = 8 spp.; Time 5 = 17 spp.; Time 6 = 26 spp.), but the quantification of the convex hull area confirmed that the stationary epifauna had the greatest extension within the morphospace. This is because some of them have developed valves with a high degree of rotation and aperture displacement (Fig 5). The highest value of the occupied area for this ecological category was recorded immediately before mass extinction, while the lowest disparity values, due to empty morphospace, were observed in the period after the event (Fig 5b). However, epifaunal disparity exhibited a relatively rapid recovery reaching values close to its maximum approximately 10 million years after the extinction event. During this time, the Mytilidae experienced an increase in species richness (from 4 to 6 spp.), presenting many similar morphologies as the now extinct Inoceramidae (Fig 6).

**Facultatively mobile infauna.** This ecological category represented the highest species richness present across all six time intervals (Time 1 = 36 spp.; Time 2 = 53 spp.; Time 3 = 8 sp.; Time 4 = 30 spp.; Time 5 = 41 spp.; Time 6 = 53 spp.). Even during the period following mass extinction (Time 3), the decline in morphological diversity was not as

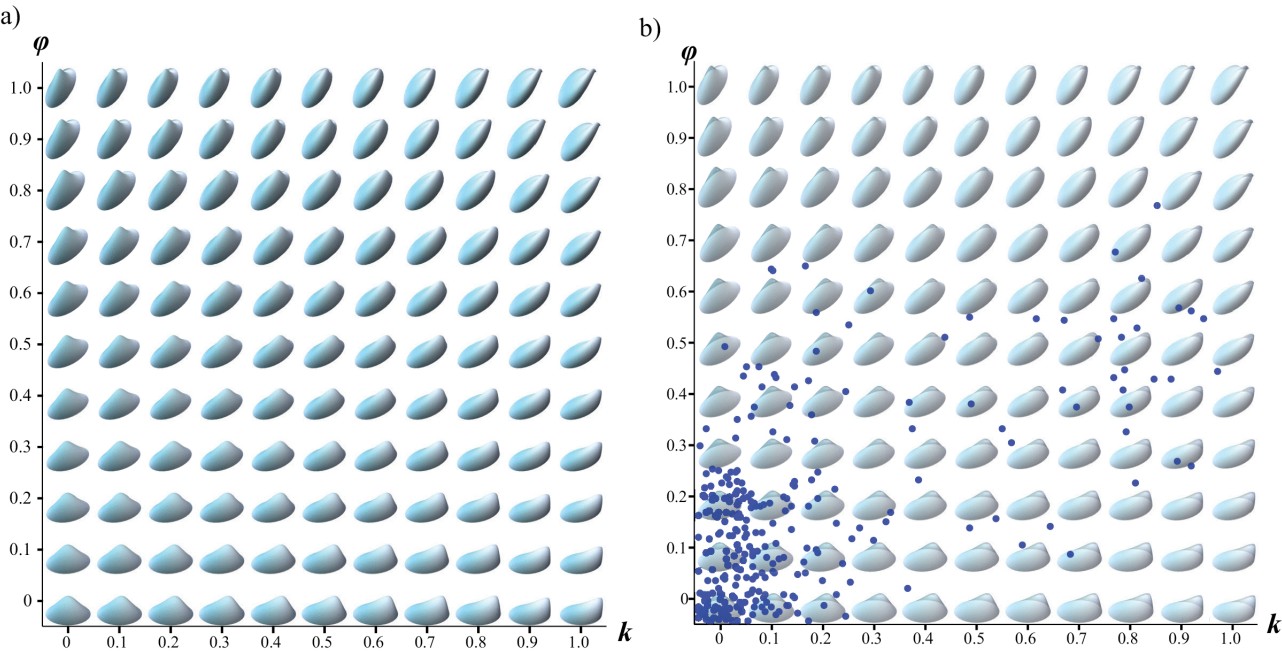

**Fig 4. a)** Theoretical morphospace generated by the systematic variation of parameters *k* and *ϕ*, and **b)** Bivalvia specimens mapped on this morphospace.

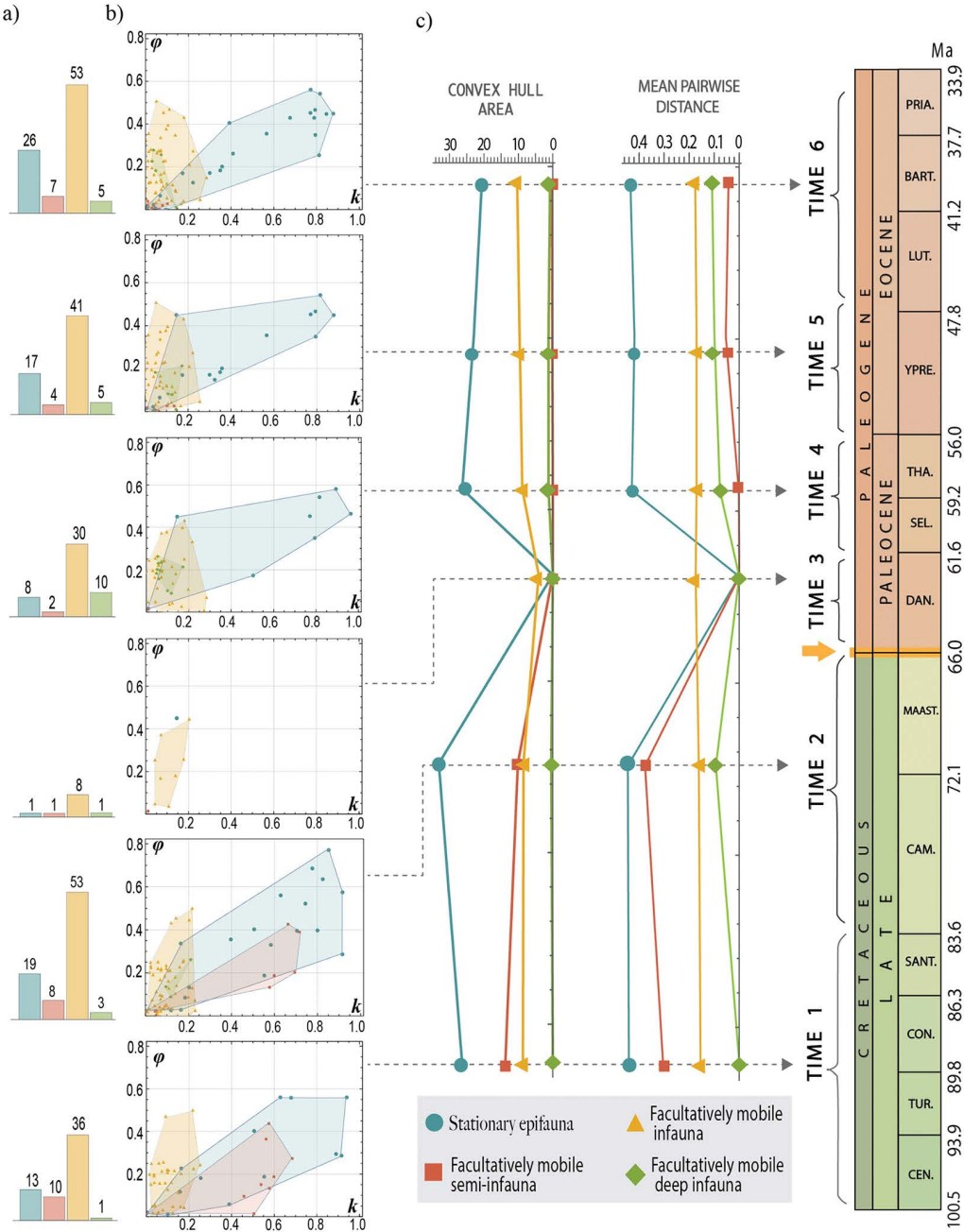

**Fig 5. Theoretical morphospaces $k-\phi$ of ecological categories of Bivalvia across the mass extinction event. a)** Histograms indicate the number of species per category. **b)** Dispersion of specimens within theoretical morphospaces across six established geological periods. **c)** Morphological disparity measures for each category and period (convex hull area and Mean Pairwise Distance). The yellow arrow indicates the K-Pg boundary.

sudden as that observed in other categories (Fig 5c). The facultatively mobile infauna developed relatively less morphological variation, particularly in the displacement of their apertures ($0<k<0.2$, Fig 5b). Most of the families present in this category persisted over time, except for Tancrediidae, which became globally extinct at the K-Pg boundary, and Astartidae, which became regionally extinct (Fig 6a). Mactridae, a species-rich, and numerically abundant family

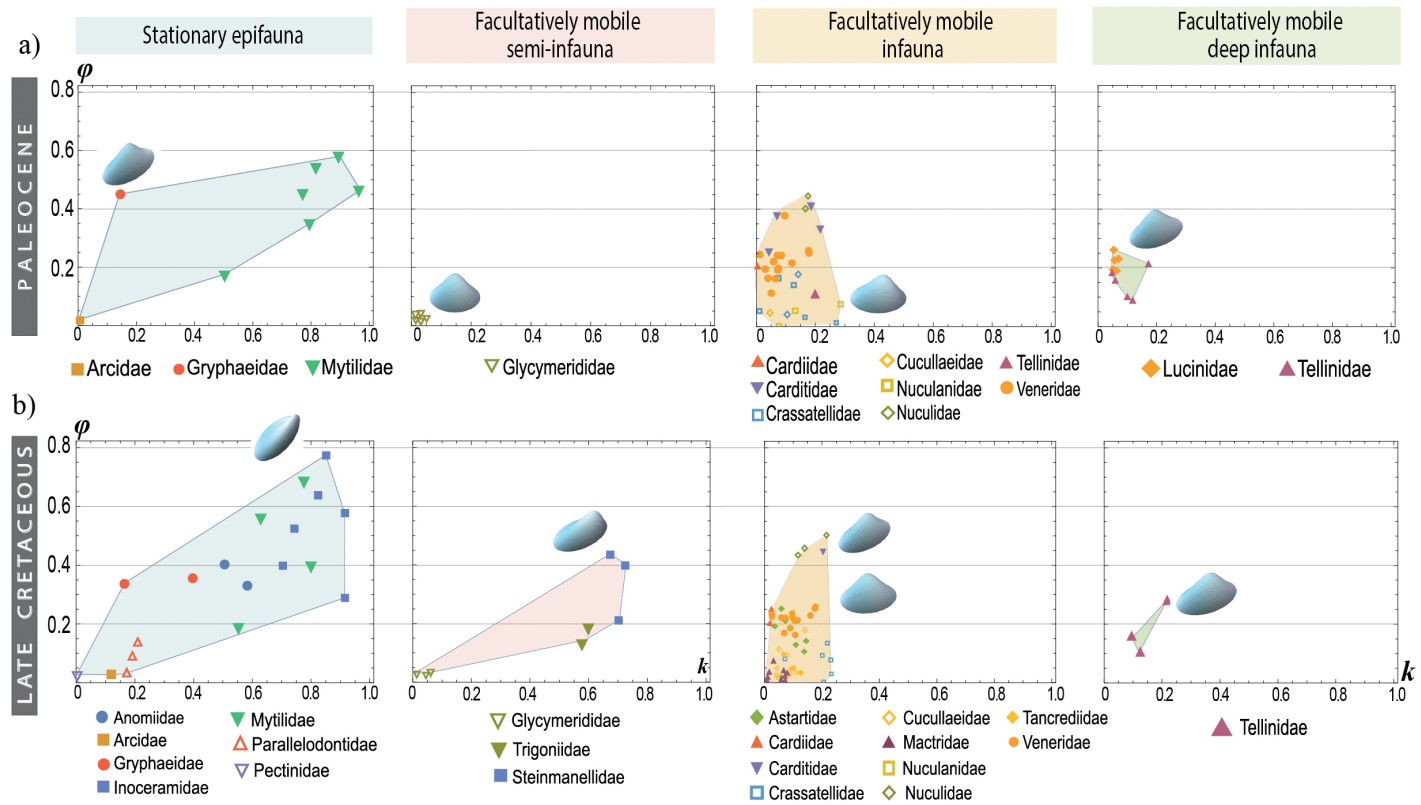

**Fig 6. Theoretical morphospaces $k-\phi$ with the existing families a) after the mass extinction (Paleocene) and b) before this event (end of the Late Cretaceous).** The four morphospaces for each period show the ecological categories of bivalves established in this study.

throughout the latest Cretaceous and throughout the early Cenozoic, were unfortunately not sampled from the Paleocene due to a lack of suitably preserved material. The family Veneridae contains the highest number of species and is mainly concentrated in an area of morphospace with almost imperceptible aperture displacement and slight inclination.

**Facultatively mobile semi-infauna.** This mode of life, along with facultatively mobile deep infauna, had the smallest occupied area in the morphospace and a smaller number of specimens (Fig 5), although these taxa are important components to Late Cretaceous and early Paleogene communities. It is important to note that semi-infaunal facultative mobile species failed to recover their morphological disparity after the K-Pg extinction, mainly because the Glycymerididae family was the only representative during the Paleogene (Fig 6). The substantial decrease in disparity is attributed to the extinction of the Trigonidae and Steinmanellidae within the Eastern Pacific (Fig 6b), which occupied a region of the morphospace very different from the Glycymerididae.

**Facultatively mobile deep infauna.** This ecological mode is represented by only one family sampled in the Cretaceous, Tellinidae, located in the same region of morphospace as the shallow infauna. It is important to highlight that after the extinction event, this ecological group experienced an increase in the number of species belonging to the Lucinidae family, which modified the dispersion of the deep infauna within the morphospace (Fig 6a), although this did not necessarily result in a larger occupied area (Fig 5c). It is worth noting that, in general, these two families differ in aperture shape, as Tellinidae tend to be more elongated. However, the bivalve morphospace was constructed using the parameters $k$ and $\phi$, which represent aperture displacement and its rotation relative to the coiling axis, respectively.

## Gastropods

The highest species density of gastropods was found in values approximately between *0.4 < a < 0.6* and *1 < z < 3* (Fig 7), which reflects a moderately elongated aperture and low vertical displacement of the spire. Morphological disparity for each category showed a notable decrease during the epoch immediately following the mass extinction (Time 3 in Fig 8c). The category that displayed the highest disparity immediately after the extinction event were gastropods that were partially buried and with locomotion capacity. According to the MPD measure, the facultatively mobile semi-infauna continued to maintain a significant disparity in all periods following the extinction event (Fig 8c). This category is exclusively represented by the species-rich Turritellidae, which has preserved its morphological diversity despite the K-Pg extinction (Fig 9).

**Creeping epifauna.** This category exhibited the highest taxonomic diversity across all six time intervals (Time 1 = 47 spp.; Time 2 = 50 spp.; Time 3 = 11 sp.; Time 4 = 33 spp.; Time 5 = 42 spp.; Time 6 = 42 spp.), and occupied the largest area within morphospace *a-z* (Fig 8). However, they also experienced the most drastic decrease in density owing to the K-Pg extinction event. The species that survived this extinction were concentrated in a specific zone of the morphospace with low *a* and *z* values, indicating morphologies with widened and narrow openings as well as a low spire height (Fig 8b). Over the 10 million years post-K-Pg boundary (Time 3), the disparity increased, but failed to fully recover. A decline in Epitoniidae species richness led to reduced morphological diversity in this ecological group, particularly in species with vertically elongated spires (*z < 4* in Fig 9).

**Creeping semi-infauna.** The Mean Pairwise Distance (MPD) values for creeping semi-infaunal gastropods were slightly higher in the latest Cretaceous (Time 2 in Fig 8c), suggesting that species in this category are somewhat more

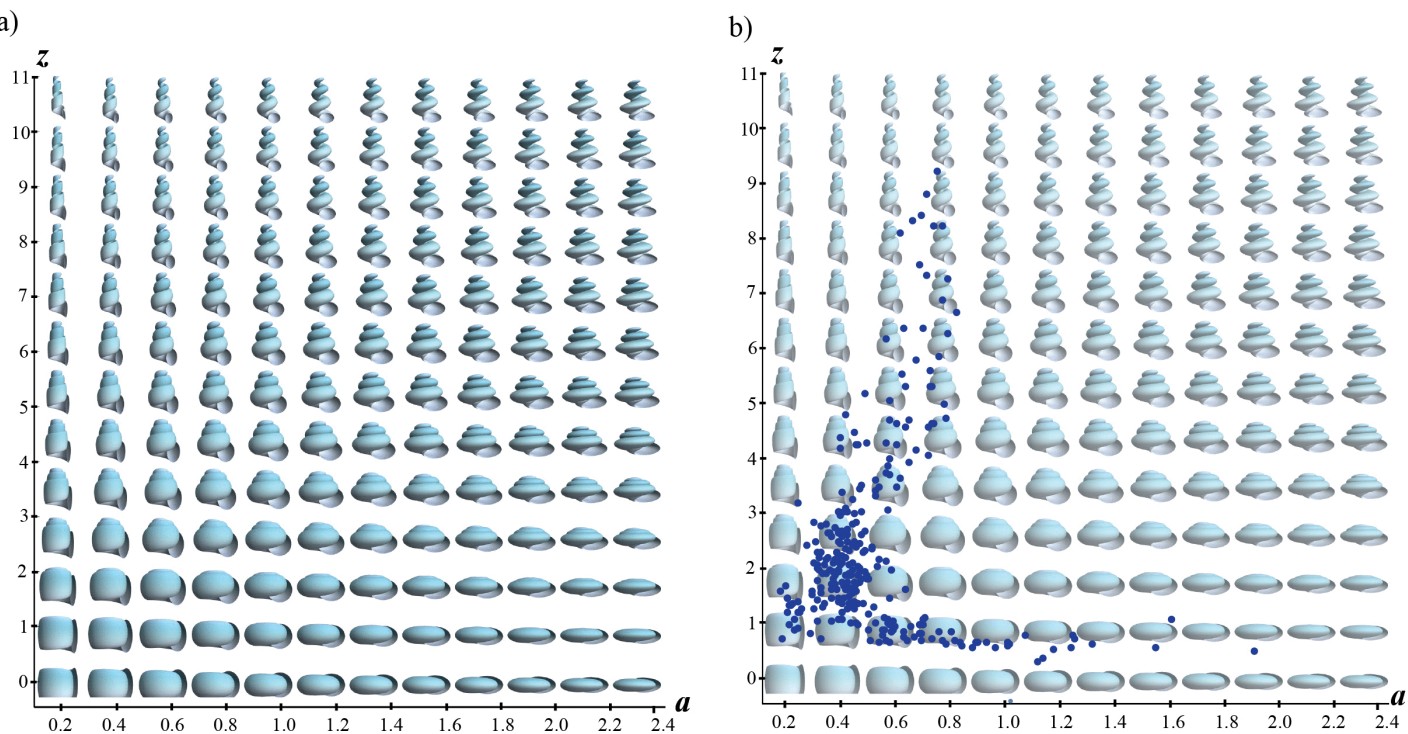

**Fig 7. a)** Theoretical morphospace generated by the systematic variation of parameters *a* and *z*, and **b)** gastropod specimens mapped on this morphospace.

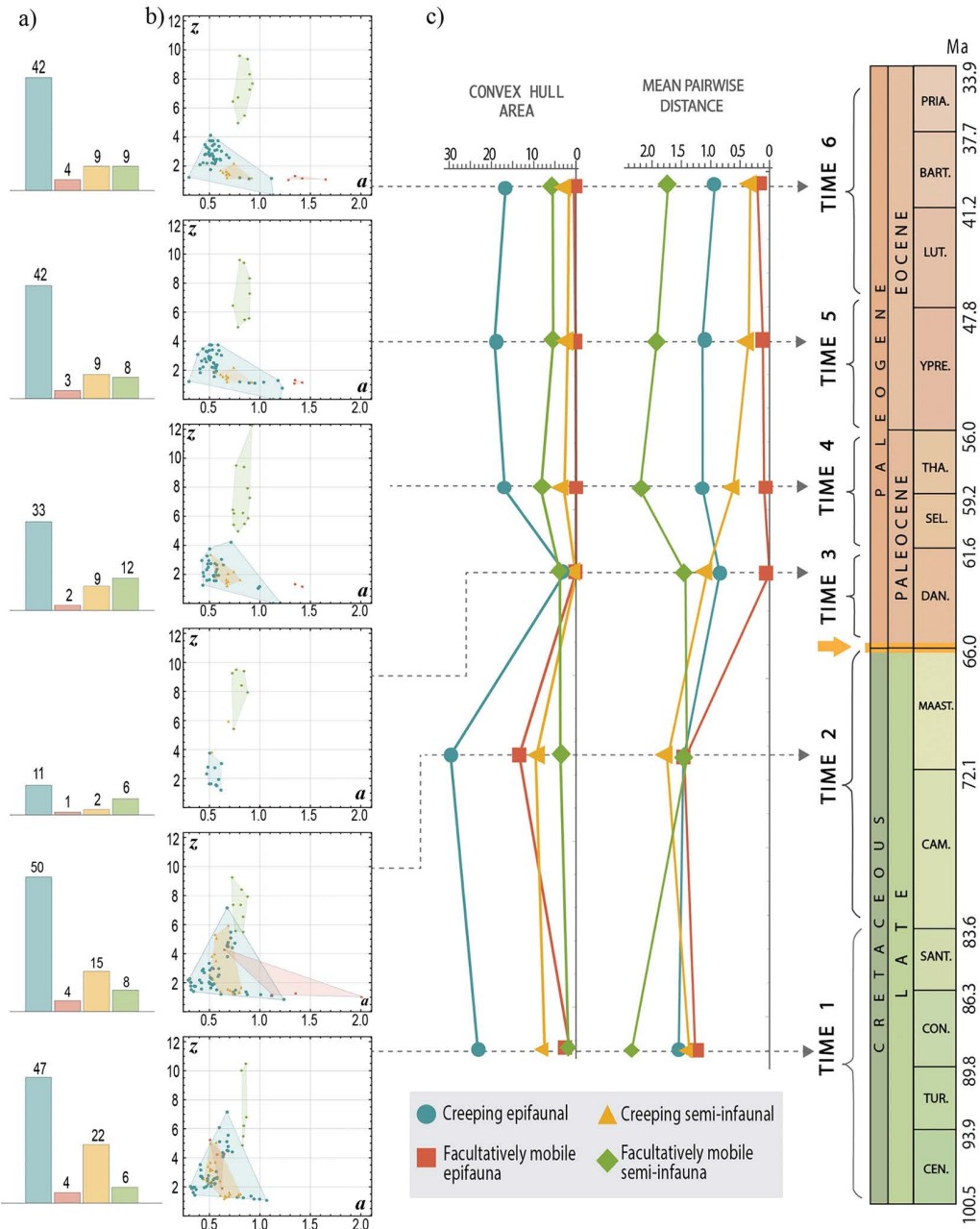

**Fig 8. Theoretical morphospaces *a–z* of ecological categories of gastropods across the mass extinction event. a)** Histograms indicate the number of species per category. **b)** Dispersion of specimens within theoretical morphospaces across six established geological periods. **c)** Morphological disparity measures for each category and period (convex hull area and Mean Pairwise Distance). The yellow arrow indicates the K-Pg boundary.

divergent from one another on average. A significant reduction was observed during the Danian Period (Fig 8, Time 3), during which only two species of the Aporrhaidae family were observed (Fig 9). The recovery of this group towards the end of the Paleocene (Selandian-Thanetian) is attributed to the diversification of species within the Naticidae (Fig 9), which spread across the morphospace with apertures ranging from elongated to circular form (*0.5 < a < 0.8*) and a slight elongation of the spire (*1 < z < 2*).

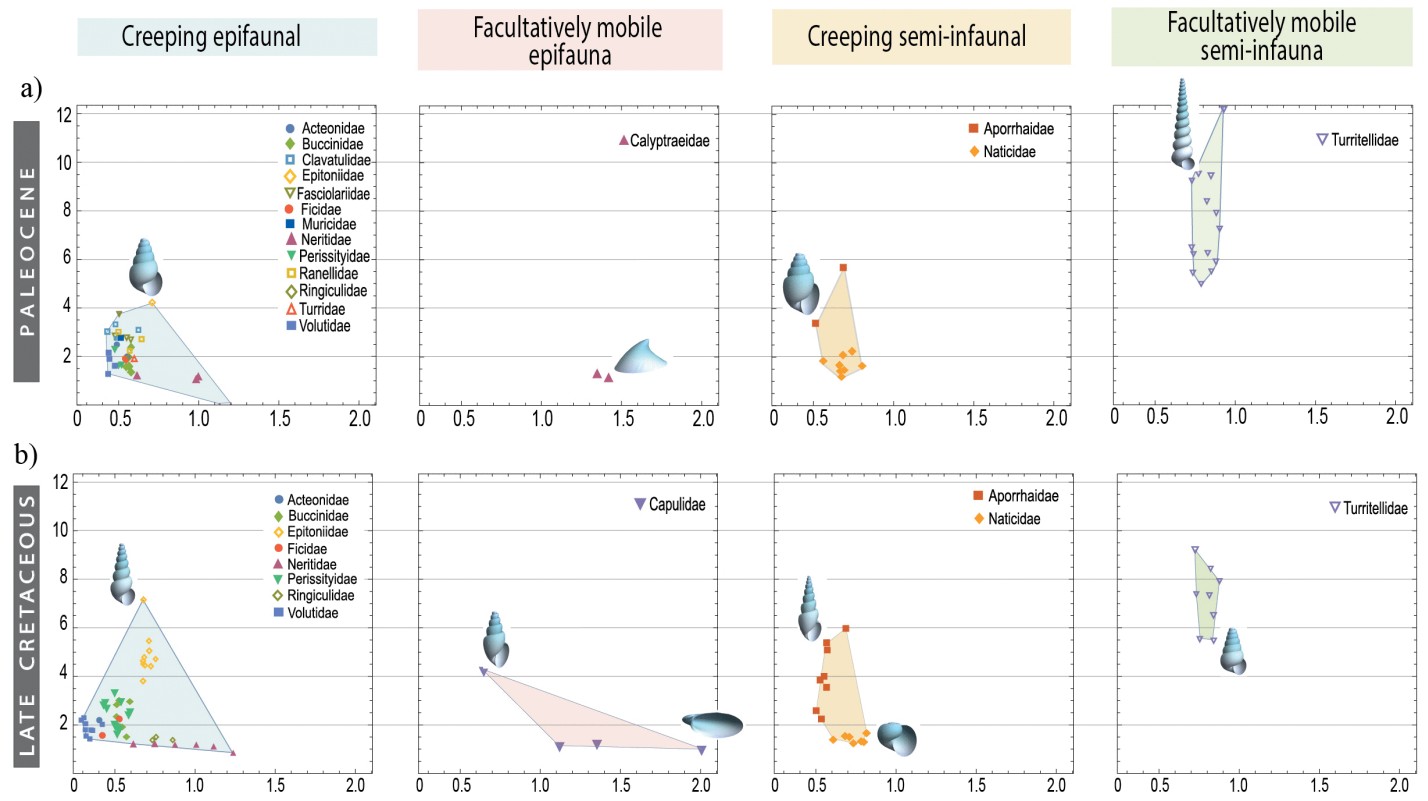

**Fig 9. Theoretical morphospaces a−z with the existing families a) after the mass extinction (Paleocene) and b) before this event (end of the Late Cretaceous).** The four morphospaces for each period show the ecological categories of gastropods established in this study.

**Facultatively mobile epifauna.** This ecological mode had the lowest number of species (from 4 to 1 spp.) and occupied the smallest area within the morphospace (Fig 8). At the beginning of the Late Cretaceous (Time 1), helicoid shells with narrow apertures emerged ($2 < z < 5$; $a \approx 0.5$; Fig 8b). By the end of the Late Cretaceous (Time 2), Capulidae species with different morphologies had expanded notably, increasing the occupied area within the morphospace (Fig 9). After the K-Pg boundary, Calyptraeidae was the only family exhibiting a facultatively mobile epifaunal ecology and remains confined to the morphospace occupied by limpet-type shells (Fig 9).

**Facultatively mobile semi-infauna.** This ecological group maintained a consistent number of species and occupied a stable region within the morphospace during the Cretaceous-Eocene period (Fig 8). It was exclusively represented by the family Turritellidae (Fig 9), indicating that the morphotype of this family experienced minimal changes over geological time. In contrast to the other groups, semi-infauna with facultative movement did not experience a significant decline at the K-Pg boundary, as observed in creeping epifauna (Fig 8c). This suggests that, despite their limited occupation of morphospace, these organisms exhibit remarkable morphological stability compared to other ecological groups.

## Discussion

In this study, the selected geometric parameters of shell morphology were applied to conduct a disparity analysis of mollusks from the North American Pacific coast across the K-Pg extinction event. Although studies exploring theoretical morphospaces remain limited, they provide a valuable framework for quantifying morphological variation using parameters directly related to shell geometry. The results obtained from the theoretical morphospaces in this study emphasize

their efficacy in analyzing macroevolutionary patterns, as they incorporate axes that define changes in shell structure and enable the measurement of morphological disparity in paleontological studies.

The mass extinction event at the K-Pg boundary is evident through suddenly unoccupied areas and distribution shifts within the morphospaces of bivalve and gastropod shells. Disparity measures across the time intervals recorded a loss of morphological diversity in all groups, associated with the mass extinction event. These disparity measures highlight that, although the number of species recovered, the zones they occupied in the morphospace did not revert to their previous configurations.

## Geometric parameters

Parameters were selected based on their ability to provide comprehensive information regarding morphological changes associated with the mode of life of each class of mollusks documented by previous authors. For bivalves, both the selected parameters ($k$ and $\phi$) are closely related to the aperture, which is the most prominent structure in their valves owing to its high expansion rate. The combination of these two parameters allowed the modeling of the expansion or reduction of the anterior and posterior areas of the valve, permitting it to be an important parameter for modeling their ability to bury or adhere to substrates [31,34,67]. Similarly, in gastropods, parameter $z$ is crucial for describing their helicoidal morphology and interpreting their ecological adaptations. In this study, the interaction between parameters $z$ and $a$ can be associated with their capacity for burrowing, substrate attachment, and facultative motility [34,39,68].

It is important to note that it was not possible to exclusively associate a particular shape with a specific ecological function, revealing the complexity of the correlation between morphology and life mode. However, there are exceptions, such as the families Turritellidae (Gastropoda) and Mytilidae (Bivalvia), which show a very restricted distribution in their morphospace. It is foolhardy to limit the ecology of an organism to only the morphology of its structures; however, this represents an initial approach to understanding the ecomorphology of the studied organisms. Future research should aim to achieve a deeper understanding of shell morphology and its ecological implications.

The model used in this study focuses on describing the general shape of mollusk shells but cannot fully represent certain anatomical features, such as the siphonal canal, outer lip, and ornamentation. However, it can be mathematically adapted to generate more realistic shells [30]. The siphonal canal, linked to chemosensation and predation, evolved multiple times and is associated with mobility, whereas its loss is related to sedentary, nonmarine habitats [69]. Excluding such features may bias ecological interpretations, particularly for carnivorous or detritivorous families (e.g., Aporrhaidae, Ranellidae, and Fasciolariidae), whose morphology is not fully captured by the current model. Another aspect that the parameters used in the current model cannot represent is shell ornamentation; however, the model has the potential to incorporate this feature in future studies. In some morphologies, shell ornamentation can function as an anchoring mechanism in excavating species, facilitating burrowing, and may also serve as a potential anti-predatory adaptation, together with the development of thicker shells [31,34]. Future ecomorphological studies should incorporate new parameters for these shell structures to improve model realism and achieve more precise paleoenvironmental reconstructions.

## Disparity measurements

In this study, we selected measures of disparity that consider both size (convex hull area) and the density of specimen distribution (MPD) in the morphospaces. These measures successfully detected changes in disparity during the established temporal intervals. It is important to use a combination of different measures as they provide more comprehensive answers to trends in disparity over time, as various authors have confirmed [11,12,70].

There are notable contrasts between the extent of morphospace occupation and the number of species. Epifaunal organisms—including both stationary bivalves and creeping gastropods—exhibit greater morphological diversity, as reflected in their broader occupation of the morphospace (Fig 5, 8). This extensive morphospace coverage does not necessarily correspond to a high species richness. In contrast, infauna and semi-infauna tend to exhibit a large number of

specimens with similar morphologies, as sediment mobility enforces similar constraints. For instance, bivalves classified as facultatively mobile infauna are restricted to a specific morphospace zone which remains relatively stable from the Late Cretaceous through the Paleogene, including the K-Pg mass extinction, primarily because most families in this category persist over time.

Additionally, Veneridae harbors the largest number of infaunal species, all of which exhibit low morphological variation, reflecting an adaptation for burrowing. This morphology is characterized by an anteroposterior elongation and an enlarged foot, which is associated with an expansion in the anterior part of the shell, facilitating vertical burrowing into the sediment. These traits correspond to an aperture inclination ($\phi$) and a slight displacement ($k$) of the aperture relative to the coiling axis.

### Ecomorphology across K-Pg boundary

Mapping specimens of the Bivalvia and Gastropoda across different geological intervals provides a valuable tool for observing the morphological changes experienced by taxonomic and ecological groups before and after the K-Pg boundary. This period includes the recovery phase, which represents the immediate stage after a mass extinction event and extends over 10 million years after the extinction event [71]; it could be less when unlithified marine sediments are studied [72]. This research reveals a similar pattern, where the recovery of morphological disparity in ecological categories is achieved during the Paleocene and remains relatively stable until the Eocene. Theoretical morphospaces indicate a loss of morphological disparity for each ecological category at the beginning of the Paleocene (Time 3), followed by a recovery 10 million years later at the end of the Paleocene (Time 4). This pattern likely reflects paleoenvironmental stabilization after the collapse of productivity and sedimentary disturbances that constrained ecological strategies during the early Paleocene. Although morphological disparity increased during the post-extinction interval, the distribution of occupied morphospace areas did not return to its pre-extinction state, as reflected in the $k$–$\varphi$ parameters for bivalves and the $a$–$z$ parameters for gastropods.

Some ecological categories fail to recover their morphological diversity, primarily because of the decline in families that previously occupied specific morphological niches. For instance, in the facultatively mobile epifaunal gastropods, the Capulidae significantly reduced their morphospace area, and their ecological niche was taken over by Calyptraeidae, which only developed limpet-like forms (Fig 9). Additionally, this study documents the recovery of morphological diversity within a different clade, where some families occupy morphospace regions previously held by extinct groups (e.g., the morphospace region or the niche occupied by Inoceramidae is now partially occupied by Mytilidae; Fig 6).

This study explored how specific morphological changes, evaluated through geometric parameters, correlate with anatomical features associated with burial and facultative movement after the K-Pg boundary (Danian). Infaunal bivalves with facultative movements possess elongated anterior regions ($\phi$) on their valves, facilitating pushing and penetration into substrates. Similarly, facultatively mobile semi-infaunal gastropods display columellar folds ($z$) and laterally flattened apertures ($a$), which aid in sediment burial.

At the global level, it is recognized that the impact of Chicxulub meteorite resulted in mechanisms that triggered a series of environmental and ecosystem changes. Acidification and suppression of primary productivity in oceans have been identified as two crucial mechanisms that contribute to marine extinction [52,73]. The results obtained in this study revealed significant ecological changes that can be discussed in the context of these mechanisms, supporting the hypothesis that acidification and loss of primary production are among the main causes of extinction in the ocean. The biological and environmental consequences of the Cretaceous–Paleogene (K-Pg) boundary were spatially heterogeneous [54,74,75]. Changes in primary production likely varied regionally, as can be discussed in relation to the ecomorphological categories observed in this study for mollusks from the North American Pacific coast.

The decrease in certain ecomorphologies in morphospace during the Danian, the immediate age following extinction, suggests ecological restructuring. Bivalves and gastropods with the highest disparity recorded in this period are capable of

burrowing into sediment (infauna or semi-infauna) and show active mobility (creeping and facultatively mobile). This agrees with previous research that documented mollusk assemblages with characteristics associated with marine environments with low nutrient levels, attributed to ocean acidification and low primary productivity at the K-Pg boundary [51,73,76–78]. Environments with low nutrient levels may intensify food search pressure. This phenomenon could have significantly contributed to the restructuring of ecological communities, where increased mobility for food searches or infaunal habits for deposit feeding proved to be successful. Sessa et al. [76] reported that, through the K-Pg boundary in the Gulf Coastal Plain, inactive suspension feeders were replaced by active suspension feeders, such as turritellid gastropods. A similar pattern is observed in this study with specimens from the Northeast Pacific Coast, where the morphological disparity of Turritellidae increased following the extinction event.

The post-extinction reduction and subsequent recovery of morphospace occupation observed in this study may reflect paleoenvironmental and sedimentary transitions across the K-Pg boundary. The collapse of primary productivity and the increase in sedimentary instability would have reduced habitat diversity and constrained ecological strategies, particularly for benthic and epifaunal taxa [79]. Based on evidence reported in previous studies, it can be hypothesized that as sedimentary conditions stabilized and nutrient availability improved during the Paleocene, ecological niches gradually reopened, allowing the recovery of both morphological disparity and functional diversity. However, the spatial distribution of the classes Bivalvia and Gastropoda within the morphospace did not return to its pre-extinction configuration.

The fossil bivalves and gastropods under examination were from the Northeast Pacific region, including localities in Baja California, California, Oregon, Washington, and British Columbia, Canada. The Late Cretaceous and early Paleogene of the Northeast Pacific were characterized by a generally warm climate, in part ameliorated by the presence of island arcs that provided protection against cold currents [59]. Consequently, the fossils studied here were sampled from deposits that offer a rich diversity of mollusk fauna associated with persistent tropical and shallow-water environments. It would be interesting for future research to compare these settings and mollusk biodiversity with those of the Western Interior Seaway, which was influenced by the transgressive and regressive processes of an epicontinental sea [80].

Given the broad geographic distribution of the fossils studied (North American Pacific coast), the results presented here can be regarded as more generalizable. Therefore, conducting a more focused study in a specific region or including additional data on paleoclimates and geographic locations could significantly enrich the results of morphological disparities.

This approach allowed for more detailed and robust results, thereby improving the discussion in terms of paleoecology. The motivation of this study, utilizing geometric parameters in disparity analysis, is further strengthened by the quantification and three-dimensional modeling of the shell morphology. These models can be subjected to functional testing, as demonstrated in recent studies [81,82].

In conclusion, the disparity measures across theoretical morphospaces from the Cretaceous to the Eocene allowed for an examination of unoccupied areas and distribution changes across the K-Pg boundary. These shifts were explored in terms of locomotion, life position, and correlation with morphological parameters that could influence the ecomorphology. The decrease in certain modes of life during the post-extinction event suggests an ecological impact that can be discussed in terms of an environment with low nutrient levels. In this scenario, organisms with infaunal morphology and active movement may be advantageous. Thus, the theoretical morphospace framework and its geometric parameters offer a practical approach for studying mollusk shell morphology during extinction events.

Future research integrating more geometric parameters may offer new perspectives for investigating macroevolutionary disparity, functional morphology, and ecomorphology within fossil mollusks. The work conducted in this study is expected to provide tools that facilitate comprehensive research incorporating theoretical morphology, highlighting the relevance of geometric models in paleontological and evolutionary research.

## Supporting information

**S1 Table. Fossil species used in this study, indicating the family, scientific name, as well as the number and name of the collections.**
(XLSX)

## Acknowledgments

Gabriela Contreras-Figueroa wishes to thank National History Museum of Los Angeles County (NHMLA) for awarding her the Student Collections Study Award and Paleontological Research Institution for PRI John W. Wells Grants-in-Aid of Research Program. Juliet Hook (NHMLA) and Leslie Skibinski (PRI) are thanked for facilitating access to specimens and data. This study benefited from collections digitized at the NHMLA through National Science Foundation (NSF) funding (NSF 1503065, 1561429, and 1902262).

## Author contributions

**Conceptualization:** Gabriela Contreras-Figueroa, Austin J.W. Hendy, José L. Aragón.

**Data curation:** Gabriela Contreras-Figueroa, Austin J.W. Hendy.

**Formal analysis:** Gabriela Contreras-Figueroa, Austin J.W. Hendy, José L. Aragón.

**Funding acquisition:** Austin J.W. Hendy.

**Investigation:** Gabriela Contreras-Figueroa, Austin J.W. Hendy, José L. Aragón.

**Methodology:** Gabriela Contreras-Figueroa, Austin J.W. Hendy, José L. Aragón.

**Resources:** Austin J.W. Hendy.

**Software:** Gabriela Contreras-Figueroa.

**Supervision:** Austin J.W. Hendy, José L. Aragón.

**Validation:** Gabriela Contreras-Figueroa, Austin J.W. Hendy, José L. Aragón.

**Visualization:** Gabriela Contreras-Figueroa.

**Writing – original draft:** Gabriela Contreras-Figueroa, José L. Aragón.

**Writing – review & editing:** Gabriela Contreras-Figueroa, Austin J.W. Hendy, José L. Aragón.

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
