## [Decision Letter · Decision Letter 0]

1 Oct 2025

Dear Dr. Aragón,

Thank you for submitting your manuscript to PLOS ONE. After careful consideration, we feel that it has merit but does not fully meet PLOS ONE’s publication criteria as it currently stands. Therefore, we invite you to submit a revised version of the manuscript that addresses the points raised during the review process.

Dear Authors,

Thank you for submitting to PLOS ONE.

After careful review, I agree with the reviewers that the manuscript requires major revisions.

Please find their comments below.

Best regards,

We look forward to receiving your revised manuscript.

Kind regards,

Alexandre Ribeiro da Silva

Academic Editor

PLOS ONE

Journal Requirements:

2. In your manuscript, please provide additional information regarding the specimens used in your study. Ensure that you have reported human remain specimen numbers and complete repository information, including museum name and geographic location.

“All necessary permits were obtained for the described study, which complied with all relevant regulations.”

“No permits were required for the described study, which complied with all relevant regulations.”

For more information on PLOS One's requirements for paleontology and archeology research, see https://journals.plos.org/plosone/s/submission-guidelines#loc-paleontology-and-archaeology-research.

[Student Collections Study Award and Paleontological Research Institution for PRI John W. Wells Grants-in-Aid of Research Program].

[Gabriela Contreras-Figueroa wishes to thank National History Museum of Los Angeles County (NHMLA) for awarding her the Student Collections Study Award and Paleontological Research Institution for PRI John W. Wells Grants-in-Aid of Research Program. Juliet Hook (NHMLA) and Leslie Skibinski (PRI) are thank for facilitating access to specimens and data. This study benefited from collections digitized at the NHMLA through National Science Foundation (NSF) funding (NSF 1503065, 1561429, and 1902262).]

[Student Collections Study Award and Paleontological Research Institution for PRI John W. Wells Grants-in-Aid of Research Program]

Reviewers' comments:

Reviewer's Responses to Questions

**Comments to the Author**

1. Is the manuscript technically sound, and do the data support the conclusions?

Reviewer #1: Yes

Reviewer #2: Partly

Reviewer #3: Partly

2. Has the statistical analysis been performed appropriately and rigorously?

Reviewer #1: N/A

Reviewer #2: No

Reviewer #3: Yes

3. Have the authors made all data underlying the findings in their manuscript fully available?

Reviewer #1: Yes

Reviewer #2: Yes

Reviewer #3: Yes

4. Is the manuscript presented in an intelligible fashion and written in standard English?

Reviewer #1: Yes

Reviewer #2: Yes

Reviewer #3: Yes

Reviewer #1: This manuscript uses a theoretical morphospace to quantify morphological disparity in bivalves and gastropods — often overlooked taxa — across the K-Pg boundary for the Northeast Pacific. The approach is interesting, the dataset is substantial, and the main claims — a marked drop in disparity immediately after the K-Pg event and partial recovery in particular ecological modes — are plausible and generally well supported by the analyses presented.

The discussion is well anchored on existing literature and provides a compelling and down-to-earth set of results. It is a challenging subject and the authors’ effort to handle it is good to the extent that some difficult categories become familiar to the readers as they progress through the manuscript. Overall, it is a scientifically sound paper that makes a meaningful contribution.

That said, I recommend accepting pending minor revisions addressed below:

1. If possible, provide better-resolution pictures of the samples depicted in Figure 3f and 3h (Caryocorbula dickersoni — LACMIP 16115 Type 6592; Corbulidae and Goniocheila wilsoni — LACMIP 17535-6; Aporrhaidae, respectively).

2. Check figures 6 and 9 and/or their captions, which say “a) before the mass extinction (end of the late Cretaceous and b) after this event (Paleogene)”, it seems to be inverted.

3. In line 209, authors state that intraspecific variation is anticipated to be minimal in the context of shape differences across families and classes, but what exactly supports this claim? Couldn’t some species display significant variations when in different microenvironments or even between individuals of different ages? Perhaps add a quantitative source or a short justification.

4. In line 236, authors mention that the increase in number of Lucinidae species does not necessarily translate into a greater extension in the morphospace, however, by looking at the graph, it seems it does, slightly.

5. I must admit that I am having a bit of a hard time observing how Mytilidae occupies the morphospace region previously held by Parallelodontidae (l. 363; Fig. 6).

6. Line 259, the figure to be cited was indeed 8c, or should it be 8b?

7. Line 266, does the creeping semi-infaunal gastropods display higher diversity of morphology during the Cretaceous period than the creeping epifaunal?

8. Lines 281-286, the figures to be cited here might be 8 and 9 instead of 7 and 8.

9. In a more general note, authors briefly remarked that certain morphological features cannot be fully replicated due to the limitations of the geometric model. Maybe add a small discussion on what this could entail – how, specifically, may it bias the interpretations made from the obtained results? Which ecological signals might be under-represented?

10. Are all the species in Table S1 indeed presented with their most recent accepted names?

11. Proofread entire manuscript for very few typos and grammar issues. The same goes for the formatting of citing figures within the text.

Reviewer #2: This study evaluated morphological diversity (disparity) changes in bivalves and gastropods across the K-Pg boundary. The authors attempt to evaluate disparity using theoretical morphological models and associated morphospaces, observing a decline in disparity with the mass extinction event near the K-Pg boundary, and subsequent recovery, and suggesting that their mode of life is related to these patterns. While these results are interesting, I cannot fully support the results and their claims in its current form for the following reasons: 1) Sample size and the need for appropriate corrections; 2) The theoretical morphological model parameters considered are likely insufficient; 3) The focal taxonomic groups are inadequate to support claims about mollusks in general; 4) Over-discussion. Thus, I conclude that substantial revision is needed before publication.

Major comments:

1. Sample size and the need for appropriate corrections

Sample size is critical for evaluating the disparity. As the size of the analyzed dataset increases, disparity metrics tend to increase accordingly. Corrections that account for sampling completeness are needed. Moreover, in Time 3, sample sizes for several life modes are reduced to 1-2 specimens, making the current evaluations among life modes fundamentally problematic. Additionally, whether the observed decrease and recovery in disparity represent meaningful changes cannot be assessed without removing the effects of taxonomic diversity decline and recovery. In other words, the results may indirectly reflect changes in taxonomic diversity through disparity metrics, rather than capturing true changes in disparity.

2. The theoretical morphological model parameters considered are likely insufficient

Of the six parameters in the theoretical morphological model (Contreras-Figueroa and Aragón 2023. Diversity) used in this study, only two parameters each are considered for bivalves (k, ϕ) and gastropods (z, a). While I agree that these two parameters for each group are important, as explained in Table 1, the exclusion of other parameters has not been adequately justified. Additional parameters are clearly utilized to reproduce the diverse morphologies shown in Figure 3. Furthermore, previous studies have demonstrated that factors corresponding to other parameters are important for understanding morphological diversity and functional constraints related to modes of life (e.g.,aperture inclination for gastropods; Raup 1966 Journal of Paleontology, Linsley 1977 Paleobiology, Noshita et al. 2012 Paleobiology, Okajima and Chiba 2013 Evolution). If only this subspace of the morphospace is to be considered, it should be explicitly stated that they are examining changes in specific aspects of disparity rather than disparity in general, and the discussion should be limited accordingly (see also Major comment 4 below).

3. The focal taxonomic groups are inadequate to support claims about mollusks in general;

Claims about mollusks in general should not be based solely on bivalves and gastropods. Even when limited to bivalves and gastropods, only isometric growth is principally considered (related to Major comment 2 above). Given that non-isometric morphological features play important roles in the life modes considered in this study, the claims of this manuscript are likely to be more limited in scope than currently presented.

4. Over-discussion

Multiple claims in the manuscript seem over-discussed to me. Given the limited size of the data, the limited sampling and geographic scope, and the limited morphological features analyzed, I recommend that the authors carefully reconsider the claims presented and clarify the scope of what can be reasonably concluded. In paleontological research, the above challenges are difficult to avoid, and I do not believe they diminish the value of the study itself. However, it is crucial to clearly describe what can be claimed based on the data and quantitative analyses. For the value of quantitative analyses using theoretical models, it is important to distinguish between what can be quantitatively supported based on hypotheses and what remains speculative.

Minor comments:

1. It would be useful for readers to provide specific methods for model parameter estimation. Since this is important for evaluating disparity, also include information on estimation errors if possible.

2. I recommend adding specimen IDs associated with the database in Table S1.

Reviewer #3: I sincerely appreciate the opportunity to review this manuscript, which I consider extremely important for the advancement of fossil morphometric studies. The topic addressed is relevant, and the data presented have great potential to contribute significantly to the field.

Overall, the work presents a solid and representative database. However, to achieve this, some aspects need further development, particularly in the Discussion section. When reading an article, the reader expects the results obtained to be contextualized and compared with the existing literature. In this regard, the current manuscript could be enriched. The data appear to be well represented, but the discussion needs to be better developed to go beyond the presentation of the results, integrating them into the already established scientific dialogue.

Another point that deserves attention is the extrapolation of interpretations. In certain sections, local or regional interpretations are replicated to explain global-scale events without further exploring the specificities of the local data themselves. Focusing more on the local implications of the results, before broadening the scope, could strengthen the study's argument.

Transparency and reproducibility are crucial to methodology, so the number of samples (n) for each temporal stratum and each location analyzed must be clearly stated. Lack of this information can introduce bias in the interpretation of the results and is crucial for it to be clearly presented so that the scientific community can accurately assess the robustness of the conclusions. I made specific comments throughout the manuscript, suggesting adjustments and refinements to the ideas presented, if the authors deem them pertinent.

This is a very important and relevant article for fossil morphometrics. I recommend its publication after reviewing the suggested modifications. I am very excited about the potential of this work and confident that, with the adjustments, it will become a valuable reference. I congratulate the authors for their effort and wish them excellent work in the final revision phase.

**Do you want your identity to be public for this peer review?** For information about this choice, including consent withdrawal, please see our Privacy Policy

Reviewer #1: No

Reviewer #2: No

Reviewer #3: No

---

## [Author Response · Author response to Decision Letter 1]

24 Nov 2025

The response of all reviewers comments are attached in the submission as a Word file.

---

## [Decision Letter · Decision Letter 1]

4 Jan 2026

Dear Dr. Aragón,

Thank you for submitting your manuscript to PLOS ONE. After careful consideration, we feel that it has merit but does not fully meet PLOS ONE’s publication criteria as it currently stands. Therefore, we invite you to submit a revised version of the manuscript that addresses the points raised during the review process.

We look forward to receiving your revised manuscript.

Kind regards,

Alexandre Ribeiro da Silva

Academic Editor

PLOS One

Journal Requirements:

Reviewers' comments:

Reviewer's Responses to Questions

**Comments to the Author**

Reviewer #1: All comments have been addressed

Reviewer #3: All comments have been addressed

2. Is the manuscript technically sound, and do the data support the conclusions?

Reviewer #1: Yes

Reviewer #3: Yes

3. Has the statistical analysis been performed appropriately and rigorously?

Reviewer #1: Yes

Reviewer #3: Yes

4. Have the authors made all data underlying the findings in their manuscript fully available?

Reviewer #1: Yes

Reviewer #3: Yes

5. Is the manuscript presented in an intelligible fashion and written in standard English?

Reviewer #1: Yes

Reviewer #3: Yes

Reviewer #1: The revised manuscript has improved substantially and reads well. The authors have addressed most previous concerns: limitations are now acknowledged, many methodological choices are better justified, and the presentation is clearer. The study remains a valuable contribution to understanding morphological disparity across the K–Pg boundary.

Aside from the changes made, on re-reading the manuscript I noticed a few additional issues that I missed in my initial review; I apologize for the oversight.

That said, I recommend acceptance pending minor revision in order to address the points listed below:

1. The corresponding author listed in the manuscript does not match the author whose e-mail is provided in the article. Please check.

2. Line 267: the cited figure for the MPD display appears to be Fig. 8c, not 8b. If it's the case, please verify and correct figure citations throughout the text.

3. In Fig. 8c the green line representing “Facultatively mobile semi-infauna” appears detached from its marker in Time 3. Is this intended? This ends up affecting the statement in l. 263-264 and l. 375: “(…) each category showed a notable decrease during the epoch immediately following the mass extinction (…)”, which wouldn’t be necessarily the case of facultatively mobile semi-infaunal gastropods.

4. Line 296: please check whether the panel showing “helicoid shells with narrow apertures” should be Fig. 8c.

5. Line 317 states that disparity measures recorded a loss of morphological diversity in all groups, associated with the mass extinction event. Is this also true for the Facultatively mobile infaunal bivalves?

6. In the legend of Figure 5, “Stationary epifauna” text is displayed with a different font than the rest of the figure.

7. Lines 342-344: This paragraph was a great addition to the manuscript, only this specific sentence about ornamentation seems a bit textually disconnected. Please consider rewording or removing that sentence for better flow.

8. Lines 357–361 repeat the claim that infaunal and semi-infaunal taxa display less morphological variation due to environmental constraints. Please combine or remove the redundancy.

9. Line 378-380 and 416: By looking at the presented data, it seems that facultatively mobile infaunal bivalves’ occupied morphospace area was able to return to its pre-extinction state.

10. Line 386: “Inoceramidae” (typo)

11. Lines 425-426: The word “here” is used twice in the sentence; one could be suppressed.

12. The ecological interpretation that burrowing and mobility were advantageous post-extinction is compelling. Consider briefly linking this to modern analogues or experimental observations that support the advantage of mobility/burrowing under environmental stress (one or two citations would strengthen the argument).

13. Throughout the text MPD is called Mean Pairwise Distance, but figure captions read “Main Pairwise Distance.” Please correct captions to Mean Pairwise Distance for consistency.

14. Check references and figure captions for minor typos and formatting. Otherwise, the manuscript appears well proofread.

Reviewer #3: All points raised were duly addressed and clarified by the authors; therefore, I recommend publication.

**Do you want your identity to be public for this peer review?** For information about this choice, including consent withdrawal, please see our Privacy Policy

Reviewer #1: No

Reviewer #3: No

---

## [Author Response · Author response to Decision Letter 2]

14 Jan 2026

Responses to all reviewer comments was included in this submission as a separate file.

---

## [Decision Letter · Decision Letter 2]

3 Feb 2026

Morphological disparity across the K-Pg boundary in mollusk shells: a theoretical morphology approach

PONE-D-25-43258R2

Dear Dr. Aragón,

We’re pleased to inform you that your manuscript has been judged scientifically suitable for publication and will be formally accepted for publication once it meets all outstanding technical requirements.

Kind regards,

Alexandre Ribeiro da Silva

Academic Editor

PLOS One

Reviewers' comments:

Reviewer's Responses to Questions

**Comments to the Author**

Reviewer #1: All comments have been addressed

2. Is the manuscript technically sound, and do the data support the conclusions?

Reviewer #1: Yes

3. Has the statistical analysis been performed appropriately and rigorously?

Reviewer #1: Yes

4. Have the authors made all data underlying the findings in their manuscript fully available?

Reviewer #1: Yes

5. Is the manuscript presented in an intelligible fashion and written in standard English?

Reviewer #1: Yes

Reviewer #1: All revisions were successfully addressed. The resulting paper is a well‑structured and thorough investigation that is a great and valuable contribution to macroevolutionary paleobiology. I recommend acceptance with no further revisions.

**Do you want your identity to be public for this peer review?** For information about this choice, including consent withdrawal, please see our Privacy Policy

Reviewer #1: No

---

## [Editor Report · Acceptance letter]

PONE-D-25-43258R2

PLOS One

Dear Dr. Aragón,

I'm pleased to inform you that your manuscript has been deemed suitable for publication in PLOS One. Congratulations! Your manuscript is now being handed over to our production team.

Kind regards,

on behalf of

Dr. Alexandre Ribeiro da Silva

Academic Editor

PLOS One